# Clinical characteristics and diagnosis of intestinal tuberculosis in clinical practice at Thailand's largest national tertiary referral center: An 11-year retrospective review

Asawin Sudcharoen[1,2], Gahwin Ruchikajorndech[1], Sitthipong Srisajjakul[3], Ananya Pongpaibul[4], Popchai Ngamskulrungroj[5], Orawan Tulyaprawat[5], Julajak Limsrivilai[1] *

1 Division of Gastroenterology, Department of Medicine, Faculty of Medicine Siriraj Hospital, Mahidol University, Bangkok, Thailand, 2 Division of Gastroenterology, Department of Internal Medicine, Faculty of Medicine, Srinakarinwirot University, Nakhon Nayok, Thailand, 3 Department of Radiology, Faculty of Medicine Siriraj Hospital, Mahidol University, Bangkok, Thailand, 4 Department of Pathology, Faculty of Medicine Siriraj Hospital, Mahidol University, Bangkok, Thailand, 5 Department of Microbiology, Faculty of Medicine Siriraj Hospital, Mahidol University, Bangkok, Thailand

* julajak.lim@mahidol.ac.th

## Abstract

### Background

Diagnosing intestinal tuberculosis (ITB) is challenging due to the low diagnostic sensitivity of current methods. This study aimed to assess the clinical characteristics and diagnosis of ITB at our tertiary referral center, and to explore improved methods of ITB diagnosis.

### Methods

This retrospective study included 177 patients diagnosed with ITB at Siriraj Hospital (Bangkok, Thailand) during 2009–2020.

### Results

The mean age was 49 years, 55.4% were male, and 42.9% were immunocompromised. Most diagnoses (108/177) were made via colonoscopy; 12 patients required more than one colonoscopy. Among those, the sensitivity of tissue acid-fast bacilli (AFB), presence of caseous necrosis, polymerase chain reaction (PCR), and culture was 40.7%, 13.9%, 25.7%, and 53.4%, respectively. Among patients with negative tissue histopathology, 4 (3.7%) and 13 (12.0%) were ITB positive on tissue PCR and culture, respectively. The overall sensitivity when all diagnostic methods were used was 63%. Seventy-six patients had stool tests for mycobacteria. The overall sensitivity of stool tests was 75.0%. However, when analyzing the 31 patients who underwent both endoscopy and stool testing, the sensitivity of stool testing when using tissue biopsy as a reference was 45.8%. Combining stool testing and tissue biopsy did not significantly increase the sensitivity compared to tissue biopsy alone (83.9% *vs.* 77.4%, respectively).

**Data Availability Statement:** All relevant data are within the paper and its Supporting Information files.

**Funding:** The authors received no specific funding for this work.

**Competing interests:** The authors have declared that no competing interests exist.

## Conclusion

Despite the availability of PCR and culture for TB, the overall diagnostic sensitivity was found to be low. The sensitivity increased when the tests were used in combination. Repeated colonoscopy may be beneficial. Adding stool mycobacteria tests did not significantly increase the diagnostic yield if endoscopy was performed, but it could be beneficial if endoscopy is unfeasible.

## Introduction

The incidence of intestinal tuberculosis (ITB) continues to increase worldwide, but the increase is more pronounced in developing countries [1–4]. ITB accounts for approximately 10% of all extrapulmonary tuberculosis and is associated with poor clinical outcomes that include intestinal obstruction, bleeding, and/or perforation [5–7]. In Thailand, tuberculosis (TB) is a major public health problem with an approximate annual incidence of 150 per 100,000 population, and 15% of cases are extrapulmonary. The clinical presentations of ITB are nonspecific and can mimic many conditions, such as Crohn's disease, Behcet's disease, and enterocolitis caused by vasculopathy or drugs [8–11]. Establishing a diagnosis of ITB is challenging because the diagnostic sensitivity of the standard diagnostic methods is low. The reported rates of diagnostic sensitivity for those methods are as follows: 17.3–31.0% for acid-fast bacilli (AFB) smear, 6–76% for tissue culture, and 21.6–65% for polymerase chain reaction (PCR) test [12]. Combining these tests or repeated colonoscopy may improve the diagnostic sensitivity, but data supporting these hypotheses remain limited [13]. Furthermore, stool tests for tuberculosis have been used in some centers but without clear supporting evidence. Moreover, significant numbers of patients require empirical anti-TB therapy based on clinical presentations, endoscopic findings, histopathologic findings, and cross-sectional imaging findings, but studies providing a comprehensive description and analysis of these findings from a single cohort of patients are scarce. Accordingly, the aim of this study was to assess the clinical characteristics and diagnosis of ITB in clinical practice at our national tertiary referral center and to explore improved methods of ITB diagnosis.

## Methods

We conducted a retrospective chart review that included 177 patients aged ≥18 years that were diagnosed with ITB at Siriraj Hospital, Mahidol University, Bangkok, Thailand, from January 2009 to December 2019. ITB was diagnosed based on at least one of the following: (1) the presence of AFB or caseous granuloma on histopathologic specimens; (2) tissue culture growing *Mycobacterium tuberculosis* (MTB); (3) tissue PCR positive for MTB; (4) evidence of intestinal lesion with active TB infection in another organ or system; and/or, (5) positive result from mycobacterial stool testing. All enrolled patients were required to have a complete clinical and/or endoscopic response to anti-TB drugs. Collected data included baseline demographic characteristics, laboratory tests, endoscopic findings, abdominal computed tomography (CT) findings, histopathologic findings, and stool mycobacterial profile (AFB stain, PCR for TB, and culture for mycobacteria). The protocol for this study was approved by the Siriraj Institutional Review Board (SIRB) on February 26th, 2020 (COA no. 166/2020). The requirement to obtain written informed consent from included patients was waived due to the anonymous retrospective nature of this study.

Baseline characteristics and clinical presentations were obtained from electronic medical records by AS and GR. Endoscopic findings were reviewed from endoscopic images by AS. Histopathologic slides and CT findings were reviewed by an experienced pathologist (AP) and radiologist (SS), respectively, using the definitions from Pulimood, *et al*. [14] and Kedia, *et al*. [15], respectively.

### Laboratory investigations for ITB

Laboratory diagnosis was performed in the Mycobacteriology and Mycology Laboratory of the Department of Microbiology, Faculty of Medicine Siriraj Hospital, Mahidol University. Fecal specimens were digested and decontaminated via the N-acetyl-L-cysteine-NaOH method [16] and centrifuged at 3,000 times gravity for 15 minutes. The decontaminated feces specimen and tissue biopsy were used for three different investigations. First, the detection of AFB was performed via auramine acid-fast staining under fluorescence microscopy [17]. Second, the specimens were subjected to direct extraction of total DNA via magnetic bead-based method using MagDEA® Dx SV reagent (Precision System Science, Chiba, Japan) according to the manufacturer's instructions. The extracted DNA was then used as template DNA for the Any-plex™ MTB/NTM Real-Time Detection assay (Seegene, Seoul, South Korea) [18]. This assay relies on real-time multiplex PCR and distinguishes between MTB and non-tuberculosis mycobacteria. Third, culturing for MTB was performed by inoculating the specimens onto egg-based (Löwenstein-Jensen; LJ) and liquid-based (Mycobacteria Growth Indicator Tube; MGIT) medium. An MTB-positive culture was first examined via auramine staining, with subsequent confirmation by real-time PCR.

Interferon-gamma release assay for tuberculosis was measured using Quantiferon-TB Gold Plus (Qiagen, Dusseldorf, Germany), a Sandwich enzyme-linked immunosorbent assay. The cutoff value was 0.35 IU/ml. The risk factors associated with the negative IGRA result include advanced age, low peripheral lymphocyte counts, and immunosuppressive conditions [19, 20].

### Statistical analysis

Descriptive statistics were used to summarize patient characteristics. Continuous variables are expressed as mean ± standard deviation or median and range as appropriate. Categorical variables are presented as numbers and percentages. A comparison of the sensitivities of tissue and stool mycobacteria tests in the same patient was performed using McNemar's test. A two-tailed *p*-value of less than 0.05 was considered significant. All analyses were performed using SAS version OnDemand for Academics (SAS Institute, Inc., Cary, North Carolina, USA).

## Results

Baseline patient characteristics are shown in Table 1. The mean age of patients was 49.0±15.9 years, and 55.4% were male. Seventy-six (42.9%) patients were immunocompromised. Pulmonary involvement was found in 77 patients (43.5%); 51 had positive sputum tests for *M. tuberculosis*. Serious complications, including intestinal perforation and gastrointestinal bleeding, were found in 25 (14.1%) and 10 (5.6%) patients, respectively.

Colonoscopy was performed in 108 cases. As shown in Table 1, the common endoscopic findings included ulcers (79.6%) and mass (16.7%). The most common ulcer characteristics were transverse (55.8%), round (41.8%), and geographic (34.9%) shape (Fig 1 and Table 1). Ulcers were most commonly observed at the right colon and ileum. Fifty-one percent, 45%, 31%, and 23% of patients had ileum, cecum, ascending colon, and transverse colon involvement, respectively, whereas only 19%, 9%, and 5% had descending, sigmoid, and rectal involvement.

**Table 1. Baseline demographic and clinical characteristics, clinical presentations, laboratory findings, colonoscopy findings, computed tomography findings, and histopathology findings in patients diagnosed with intestinal tuberculosis.**

| Demographic and clinical characteristics | N = 177 |
|---|---|
| Age (years), mean±SD | 49.0±15.9 |
| Male gender, n (%) | 98 (55.4%) |
| Diabetes mellitus, n (%) | 19 (10.7%) |
| Chronic kidney disease, n (%) | 15 (8.5%) |
| Cirrhosis, n (%) | 7 (3.9%) |
| Immunocompromised, n (%) | 76 (42.9%) |
| • Human immunodeficiency virus infection | 55 (31.1%) |
| Median CD4 count (cells/mm$^3$) | 79 (range 3–880) |
| Received antiretroviral therapy | 10/55 (18.2%) |
| • Corticosteroids | 18 (10.2%) |
| • Chemotherapy | 2 (1.1%) |
| • Other immunosuppressive agents | 15 (8.5%) |
| **Clinical presentations, n (%)** | |
| Mean duration from symptom onset to hospital presentation (months) | 5.5 |
| Gastrointestinal symptoms | |
| • Abdominal pain | 92 (52.0%) |
| • Diarrhea | 78 (44.1%) |
| • Lower gastrointestinal bleeding | 25 (14.1%) |
| • Obstruction | 19 (10.7%) |
| • Abdominal mass | 14 (7.9%) |
| • Perforation | 10 (5.6%) |
| • Perianal lesion | 5 (2.8%) |
| • Ascites | 3 (1.7%) |
| Systemic symptoms | |
| • Weight loss | 93 (52.5%) |
| • Fever | 79 (44.9%) |
| • Night sweats | 8 (4.5%) |
| Pulmonary involvement | 77 (43.5%) |
| Lymphadenopathy | 16 (9.0%) |
| **Laboratory findings** | |
| Hemoglobin (g/dL), mean±SD | 10.4±2.7 |
| Platelet (*10$^3$/mm$^3$), mean±SD | 307±140 |
| Albumin (g/dL), mean±SD | 3.31±0.84 |
| Interferon-gamma release assay, n (%) | 8/9 (88.9%) |
| **Colonoscopy findings** | **(n = 108)** |
| Ulcer, n (%) | 86 (79.6%) |
| Size (cm), mean±SD | 1.6±1.7 |
| Number of ulcers, n (%) | |
| • Single ulcer | 28 (32.6%) |
| • Multiple ulcers | 52 (60.4%) |
| • Unknown | 6 (7.0%) |
| Ulcer characteristics, n (%) | |
| • Transverse | 48 (55.8%) |
| • Round | 36 (41.8%) |
| • Geographic | 30 (34.9%) |
| • Aphthous | 12 (13.9%) |

*(Continued)*

**Table 1.** (Continued)

| Demographic and clinical characteristics | N = 177 |
|---|---|
| • Longitudinal | 2 (2.3%) |
| Mucosal nodularity, n (%) | 11 (10.2%) |
| Stricture, n (%) | 10 (9.3%) |
| Mass, n (%) | 18 (16.7%) |
| **Computed tomography findings, n (%)** | **(n = 75)** |
| Bowel wall findings | |
| • Bowel wall thickening | 59 (78.7%) |
| ○ Symmetrical wall thickening | 46/59 (78.0%) |
| ○ Asymmetrical wall thickening | 13/59 (22.0%) |
| • Bowel hyperenhancement | 43 (57.3%) |
| • Bowel wall stratification | 35 (46.7%) |
| • Target sign | 16 (21.3%) |
| Involved pattern | |
| • Long segment involvement | 45 (60.0%) |
| • Multiple segment involvement | 15 (20.0%) |
| Mesenteric findings | |
| • Fibrofatty proliferation | 45 (60.0%) |
| • Comb sign | 10 (13.3%) |
| Abdominal lymphadenopathy | 46 (61.3%) |
| • Necrotic lymph node | 26/46 (56.5%) |
| • Calcified lymph node | 2/46 (4.3%) |
| • Enlarged lymph node without necrosis and calcification | 18/46 (39.2%) |
| Ascites | 20 (26.7%) |
| **Histopathology findings, n (%)** | **(n = 51)** |
| Granuloma | 40 (78.4%) |
| Confluent granuloma | 30 (58.8%) |
| Mucosal granuloma | 37 (72.5%) |
| Submucosal granuloma | 26 (51.0%) |

SD, standard deviation

Seventy-five patients underwent abdominal CT scans. The most common abnormal findings included bowel wall thickening (78.7%) and fibrofatty proliferation (60%). Abdominal lymphadenopathy was found in 46 patients (61.3%), among which 26 (56.5%) had nodal necrosis. Histopathologic specimens were available for review in 51 patients because some specimens were not kept, and some specimens had the stain fading out, resulting in an inability to be reviewed. Forty patients had epithelioid granuloma, of which 30 (58.8%) were described as confluent. (Table 1).

## Diagnosis of intestinal tuberculosis by colonoscopy

As shown in Fig 2, most study patients (108/177, 61.0%) underwent colonoscopy. Twenty-four (13.6%) patients had complicated diseases at the presentation that required surgery, and 45 (25.4%) patients were diagnosed based on clinical and stool tests without endoscopy. Among the 108 patients who underwent colonoscopy, specimens were sent for histopathologic analysis in 108 patients, PCR for TB in 74 patients, and mycobacterial culture in 73 patients. The mean number of biopsies sent for histopathologic analysis was 8.6±5.4 specimens per colonoscopy.

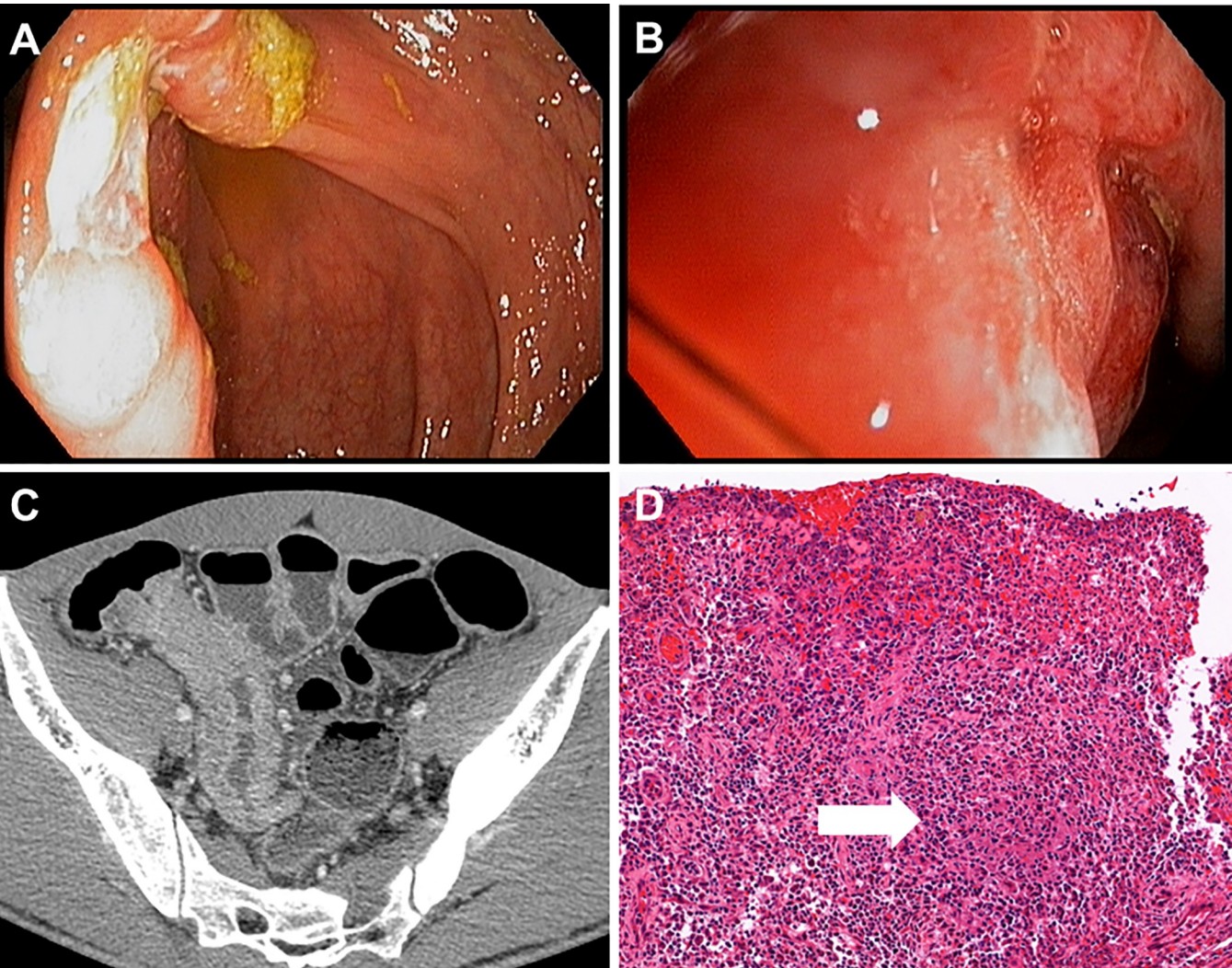

**Fig 1.** Endoscopic (A and B), computed tomography (C), and pathological findings (D) from the patient with intestinal tuberculosis. A and B showed a large ulcer involving the IC valve. C was an axial contrast-enhanced CT scan showing moderate segmental wall thickening with increased mucosal and mural enhancement at the terminal ileum with associated increasing IC valve thickness. D showed ulcerated ileal mucosa with dense chronic inflammatory cell infiltration and aggregates of epithelioid histiocytes (granuloma) with central necrosis (arrow).

As shown in Table 2, the sensitivity of tissue AFB, caseous necrosis, and pathology (presence of either AFB or caseous necrosis) was 40.7% (44/108), 13.9% (15/108), and 47.2% (51/108), respectively. The sensitivity of PCR for TB and mycobacterial culture was 25.7% (19/74) and 53.4% (39/73), respectively. The overall sensitivity of the combination of all tests was 63.0% (68/108). As shown in Fig 3, among the 57 (52.8%) patients with negative tissue pathology, 14 (13.0%) were diagnosed based on either PCR or culture; 1 (0.9%) was diagnosed by only PCR for TB; 10 (9.2%) were diagnosed by only mycobacterial culture; and, 3 (2.7%) were diagnosed by positive PCR and positive culture.

Fig 3 shows the number of patients diagnosed by each diagnostic modality at each of up to four colonoscopies. Fig 3 also shows the empiric treatment given (TB elsewhere/no TB elsewhere) at each of the 4 colonoscopy time points. Ninety-six (88.9%) patients underwent colonoscopy one time, 9 (8.3%) patients required two colonoscopies, and 3 (2.7%) patients

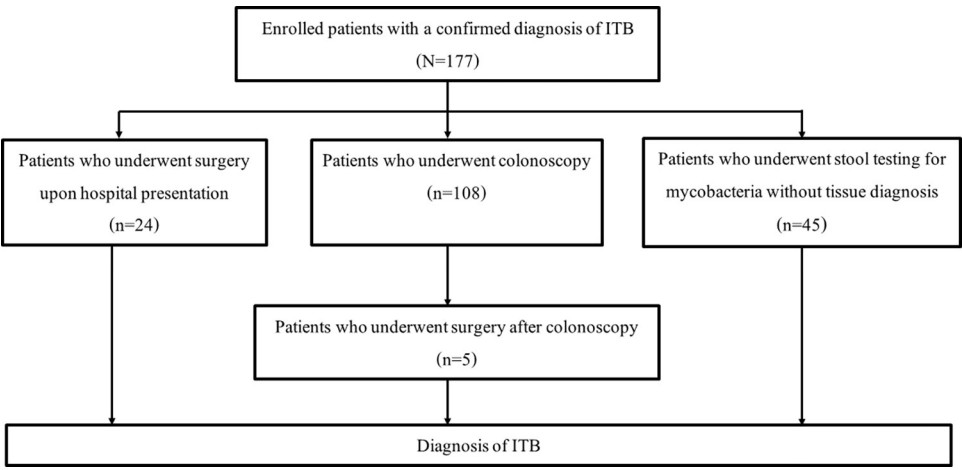

**Fig 2. Diagnostic methods of intestinal tuberculosis of patients.**

required four colonoscopies. Among the 9 patients who benefited from one repeat colonoscopies, 4 had positive findings from histopathologic specimens, 5 had *M. tuberculosis* growth on mycobacterial culture, and no patients had positive PCR results. However, in the 3 patients who underwent colonoscopy more than two times, only one patient benefited from tissue biopsy, which was a positive mycobacterial culture. In this patient, culture for mycobacteria was not sent after the first three colonoscopies.

## Diagnosis of intestinal tuberculosis via stool mycobacterial profiles

Seventy-six patients had a stool test for mycobacteria. Of those, 59 (77.6%) patients were immunocompromised, with the predominant cause being HIV infection (49 patients, 59.2%). Of 76 patients, 45 had only a stool test, and 31 had both a stool test and a colonoscopy. As shown in Table 2, stools were sent for AFB stain in 75 patients, PCR for TB in 48 patients, and mycobacterial culture in 64 patients. The sensitivities of the stool tests were 52.0% (39/75) for AFB staining, 64.6% (31/48) for PCR for TB, 62.5% (40/64) for mycobacterial culture, and 75.0% (57/76) for the combination of all stool tests.

**Table 2. The sensitivity of each diagnostic modality used to evaluate tissue biopsies obtained by colonoscopy and stool specimens for mycobacteria.**

| Diagnostic modality | Number of patients tested | Sensitivity |
|---|---|---|
| **Colonoscopy** | | |
| • Acid-fast stain | 108 | 44 (40.7%) |
| • Caseous necrosis | 108 | 15 (13.9%) |
| • Pathology (either acid-fast bacilli or caseous necrosis) | 108 | 51 (47.2%) |
| • Polymerase chain reaction for tuberculosis | 74 | 19 (25.7%) |
| • Mycobacterial culture | 73 | 39 (53.4%) |
| • Combination of all tests | 108 | 68 (63.0%) |
| **Stool testing** | | |
| • Acid-fast stain | 75 | 39 (52.0%) |
| • Polymerase chain reaction for tuberculosis | 48 | 31 (64.6%) |
| • Mycobacterial culture | 64 | 40 (62.5%) |
| • Combination of all tests | 76 | 57 (75.0%) |

## Patients who underwent colonoscopy
### (N=108)

**Diagnosis at 1st colonoscopy**
(n=96)

| | |
|---|---|
| **Positive histopathology** | (n=47) |
| **Among patients with negative pathology** | |
| • Positive PCR alone | (n=1) |
| • Positive culture alone | (n=5) |
| • Positive PCR and positive culture | (n=3) |
| **Empiric treatment** | (n=37) |
| • TB elsewhere | (n=16) |
| • No TB elsewhere | (n=21) |

**Diagnosis at 2nd colonoscopy**
(n=9)

| | |
|---|---|
| **Positive histopathology** | (n=4) |
| **Among patients with negative pathology** | |
| • Positive PCR alone | (n=0) |
| • Positive culture alone | (n=4) |
| • Positive PCR and positive culture | (n=0) |
| **Empiric treatment** | (n=1) |
| • TB elsewhere | (n=0) |
| • No TB elsewhere | (n=1) |

**Diagnosis at 3rd colonoscopy**
(n=0)

**Diagnosis at 4th colonoscopy**
(n=3)

| | |
|---|---|
| **Positive histopathology** | (n=0) |
| **Among patients with negative pathology** | |
| • Positive PCR alone | (n=0) |
| • Positive culture alone | (n=1) |
| • Positive PCR and positive culture | (n=0) |
| **Empiric treatment** | (n=2) |
| • TB elsewhere | (n=0) |
| • No TB elsewhere | (n=2) |

**Fig 3. Diagnostic methods in 108 patients undergoing colonoscopy.**

**Table 3. Stool test results compared to tissue biopsy results in patients who underwent both stool testing for mycobacteria and evaluation of tissue biopsies obtained during colonoscopy to diagnose intestinal tuberculosis (N = 31).**

|  | Tissue positive (n = 24) | Tissue negative (n = 7) |
|---|---|---|
| Stool positive (n = 13) | 11 (35.5%) | 2 (6.5%) |
| Stool negative (n = 18) | 13 (41.9%) | 5 (16.1%) |

Among the 31 patients that had both stool testing and colonoscopy to diagnose ITB, the stool test results compared to the tissue biopsy results are shown in Table 3. The sensitivity of stool testing when using tissue biopsy as a standard reference was 45.8%. There were 13 patients who had positive tissue biopsy, but negative stool tests. Only two patients had positive stool tests, but negative tissue biopsy. Using a combination of both stool testing and tissue biopsy from colonoscopy did not significantly increase the sensitivity of ITB diagnosis compared to tissue biopsy alone (83.9% *vs.*77.4%, $p = 0.5$).

## Discussion

ITB is a relatively common condition in Asia. The symptoms can mimic various disorders of the intestine, particularly Crohn's disease [5, 7, 10]. Although many diagnostic tests are available, including histopathologic examination, PCR for TB, and mycobacterial culture from intestinal tissue, the diagnosis of ITB remains challenging. The sensitivity of each of these tests is limited due to the paucibacillary nature of the bacteria, so various combinations of these tests are commonly used in clinical practice to increase the sensitivity for detecting ITB. Furthermore, in patients with negative diagnostic tests but sustained clinical suspicion, response to empirical treatment with anti-tuberculosis agents is also used to establish the diagnosis [12, 21, 22].

In this report, we describe the clinical manifestations, endoscopic, histopathologic, and CT findings of 177 patients diagnosed with ITB, and 76 were immunocompromised. The mean age of patients was 49 years, and there was no gender predominance. The mean duration from symptom onset to hospital presentation was 5.5 months. Most patients had nonspecific symptoms, including weight loss, abdominal pain, and fever in approximately half of the patients. Lower gastrointestinal bleeding was presented in only 14% of patients. These findings are comparable to many cohorts from Asia [23–28]. Pulmonary involvement was found in 43% of patients, while only 3% had perianal diseases. These two findings help differentiate ITB from Crohn's disease, which is an important differential diagnosis [29]. The reported prevalence of pulmonary involvement ranges from 13% to 67% in previous reports [30]. Interestingly, 24 (14%) patients had complicated diseases that required surgery at the initial presentation, and another 5 (3%) patients required an operation after the diagnosis was made; thus, an overall surgical treatment rate of 17%. The reported surgery rates in other cohorts ranged from 15.0% to 43.5% [31–35].

Colonoscopy plays an essential role in the diagnosis of ITB. Suggestive endoscopic features include transverse ulcers, mucosal nodularity, and deformed ileocecal valve [11, 34, 36–38]. In our study, ulcers were observed in 79.6% of cases, and about 60% of those had multiple ulcers. However, only 55.8% had transverse ulcers, while the remaining ulcer characters were nonspecific. Similar to other cohorts, ITB predominantly involved the terminal ileum and right colon, and less than 10% of patients had rectosigmoid involvement [28]. Although endoscopic features can yield some clues for the diagnosis, definite diagnosis still requires tissue biopsy. In our cohort, the overall sensitivity was 40.7% for AFB, 13.9% for caseous necrosis, 44.2% for histopathologic diagnosis (including either positive AFB or caseous necrosis), 25.7% for PCR

for TB, and 53.4% for mycobacterial culture. The sensitivity of AFB stain has been variously reported, ranging from 0%-59% [23, 39–45]. This variation may be due to a difference in the number of biopsies. Kim, *et al.* reported a series of 42 ITB patients who underwent colonoscopy. The number of tissue biopsies ranged from 2 to 8 specimens, and an increased number of biopsies correlated with increased positivity of AFB staining [46]. Therefore, experts recommend taking at least 8 biopsy specimens during colonoscopy to diagnose ITB [47]. Our study had a mean number of biopsy specimens of 8.6 pieces, which may explain the relatively high sensitivity of AFB stain in our study cohort.

The sensitivity for diagnosing ITB by mycobacterial culture may depend on the type of culture medium [21]. The conventional method using Lowenstein-Jensen medium (LJM) has poor sensitivity ranging from 6–48%. In contrast, the mycobacterium growth indicator tube (MGIT) system (BACTEC), which facilitates early detection of mycobacterial growth, demonstrated higher sensitivity of up to 76% [36, 48]. Both of these culture systems were used in our study as standard culture methods. We reported a sensitivity of 53.4% in this cohort, which is consistent with the findings of previous studies. Interestingly and importantly, tissue culture was the only method that could successfully detect *M. tuberculosis* in 10 patients (9%). Although tissue culture for mycobacteria is helpful, its main drawback is that it requires an extended period of time for *M. tuberculosis* to grow. Tissue PCR is another commonly used diagnostic modality that has a shorter turnaround time compared to tissue culture. However, the sensitivity of tissue PCR was only 25.7% in this study, which is relatively low compared to other studies that reported rates of sensitivity ranging from 21.6% to 65% [7, 21, 49]. Only one patient in this cohort was diagnosed based on tissue PCR only.

The combination of tissue histopathology, tissue PCR for TB, and culture for mycobacteria should increase the sensitivity for diagnosing ITB. In this cohort, the sensitivity increased from 44.2% when only histopathology was used to 63.0% when all diagnostic modalities were used in combination. Many studies have reported the benefit of combining diagnostic modalities, particularly when combining histopathology with mycobacterial culture. In those studies, the reported sensitivity increased from 13–80% for histopathologic analysis alone to 75–92% when the histopathologic analysis was combined with mycobacterial culture [23, 33, 36, 40, 41, 50]. Repeated colonoscopy with tissue biopsy may also confer added benefits. Our cohort showed that nine of 12 patients who underwent repeat colonoscopy realized added benefits. The benefit of repeat colonoscopy has not been previously reported. We hypothesize that repeat colonoscopies were performed as the disease progressed, which means that the mycobacterial load in colonic tissue would have been higher, leading to a higher probability of organism detection. Although repeat colonoscopy may be beneficial, colonoscopy should not be performed more than two times. None of the third and fourth colonoscopy further detected the organism except one whose tissue samples were not sent for mycobacterial culture in his first three colonoscopies.

We also retrospectively reviewed the diagnosis of ITB via stool-based tests, including fecal AFB stain, PCR for TB, and culture for mycobacteria. Stool tests may be helpful when colonoscopy cannot be performed. Moreover, stool testing may improve the diagnostic yield when combined with colonoscopy, but the published evidence to support this hypothesis remain scarce. Sekine, *et al.* reported the sensitivities of stool AFB stain, PCR for TB, and culture for mycobacteria of 37%, 23%, and 47%, respectively [44]. In our cohort, stool AFB stain, PCR for TB, and culture for mycobacteria yielded diagnostic sensitivities of 52%, 64%, and 62%, respectively, and the sensitivity increased to 75% when all stool tests were used in combination. Any enthusiasm enlivened by this 75% rate of diagnostic sensitivity should be tempered by the existential presence of patient selection bias. By way of explanation, the patients whose stools were tested in our cohort were mainly patients with symptomatic HIV infection that were at high

risk for TB infection. Furthermore, the diagnosis of ITB was made by stool test alone in many patients. When we analyzed only patients who had both stool testing and colonoscopy performed, the sensitivity of stool testing was only 45.8% when using tissue biopsy as a standard reference. Furthermore, the diagnostic yield did not statistically significantly increase when stool tests were added to colonoscopy. Therefore, stool tests for mycobacteria may benefit only specific groups of patients–particularly when a colonoscopy cannot be performed. Although this study profoundly narrows the knowledge gap, continued study is needed to improve our understanding of the benefits and limitations of existing diagnostic methods, and to stimulate exploration of enhanced methods of diagnosing ITB.

## Strengths and limitations

Our study has many notable strengths. First, this study included a substantial number of patients. Second, we had the enrolled histopathologic and CT findings reviewed by an expert pathologist and expert radiologist, respectively, according to the definitions in the published literature. Third, we reported the benefit of repeat colonoscopy and stool test diagnostic performance, which has only rarely been reported. Fourth and last, the results of this study reflect real-world clinical practice.

This study also has some mentionable limitations. First, our study's retrospective design prohibited us from collecting all relevant diagnostic modalities from all participants. The diagnostic tests were sent depending on the treating physicians' individual decision and the tests' availability at that time. Second, patients without TB infections were not included in the cohort, so the specificity and false positive results could not be evaluated. Third, data were collected from a single center, which also happens to be a large, urban, university-based national tertiary referral center. As such, our findings may not be immediately generalizable to other care settings or levels of care. Fourth and finally, we lacked data regarding new diagnostic tests, such as interferon-gamma release assay (IGRA), which may improve the diagnosis ITB. Further study in the diagnostic efficacy of IGRA in ITB is, therefore, warranted.

## Conclusion

Despite the availability of PCR and culture for TB, the overall diagnostic sensitivity was found to be low in this cohort. The sensitivity increased when these tests were used in combination. Repeated colonoscopy may be beneficial. Adding stool mycobacteria tests did not significantly increase the diagnostic yield if endoscopy was performed, but could be beneficial if endoscopy is unfeasible.

## Supporting information

**S1 Data.**
(XLSX)

## Acknowledgments

The authors gratefully acknowledge Assistant professor Keven Jones for language editing of this manuscript.

## Author Contributions

**Conceptualization:** Gahwin Ruchikajorndech, Julajak Limsrivilai.

**Data curation:** Asawin Sudcharoen, Gahwin Ruchikajorndech, Sitthipong Srisajjakul, Ananya Pongpaibul, Popchai Ngamskulrungroj, Orawan Tulyaprawat, Julajak Limsrivilai.

**Formal analysis:** Gahwin Ruchikajorndech, Julajak Limsrivilai.

**Investigation:** Asawin Sudcharoen, Gahwin Ruchikajorndech, Sitthipong Srisajjakul, Ananya Pongpaibul, Popchai Ngamskulrungroj, Orawan Tulyaprawat.

**Methodology:** Julajak Limsrivilai.

**Project administration:** Gahwin Ruchikajorndech, Julajak Limsrivilai.

**Resources:** Sitthipong Srisajjakul, Ananya Pongpaibul, Popchai Ngamskulrungroj, Orawan Tulyaprawat.

**Supervision:** Julajak Limsrivilai.

**Validation:** Julajak Limsrivilai.

**Visualization:** Asawin Sudcharoen, Gahwin Ruchikajorndech, Julajak Limsrivilai.

**Writing – original draft:** Asawin Sudcharoen, Gahwin Ruchikajorndech.

**Writing – review & editing:** Asawin Sudcharoen, Sitthipong Srisajjakul, Ananya Pongpaibul, Popchai Ngamskulrungroj, Julajak Limsrivilai.

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
