## [Decision Letter · Decision Letter 0]

21 Nov 2022

PONE-D-22-30114Clinical characteristics and diagnosis of intestinal tuberculosis in clinical practice at Thailand’s largest national tertiary referral center: An 11-year retrospective reviewPLOS ONE

Dear Dr. Limsrivilai,

Thank you for submitting your manuscript to PLOS ONE. After careful consideration, we feel that it has merit but does not fully meet PLOS ONE’s publication criteria as it currently stands. Therefore, we invite you to submit a revised version of the manuscript that addresses the points raised during the review process.

We look forward to receiving your revised manuscript.

Kind regards,

Mao-Shui Wang

Academic Editor

PLOS ONE

Journal Requirements:

Reviewers' comments:

Reviewer's Responses to Questions

**Comments to the Author**

1. Is the manuscript technically sound, and do the data support the conclusions?

Reviewer #1: No

Reviewer #2: Yes

Reviewer #3: No

2. Has the statistical analysis been performed appropriately and rigorously? 

Reviewer #1: No

Reviewer #2: I Don't Know

Reviewer #3: No

3. Have the authors made all data underlying the findings in their manuscript fully available?

Reviewer #1: No

Reviewer #2: Yes

Reviewer #3: Yes

4. Is the manuscript presented in an intelligible fashion and written in standard English?

Reviewer #1: Yes

Reviewer #2: Yes

Reviewer #3: Yes

5. Review Comments to the Author

Reviewer #1: 1. Why were patients with perforation of tuberculous intestinal ulcers and intestinal obstruction included in the study?

2. How many patients with pulmonary tuberculosis were found to have MBT in sputum?

3. What was the level of CD-lymphocytes in patients with HIV infection? How many of them received antiretroviral therapy?

4. Figure 1C does not respond to intestinal tuberculosis. More consistent with Crohn's disease. What phase of the statement is shown in the figure?

5. The statement that abdominal lymphadenopathy in patients amounted to 611.3% (61,3% mistake in main document) is questionable.

6. In table 2, the sensitivity is calculated incorrectly. These are just descriptive statistics.

Reviewer #2: The authors report a single institution retrospective review of clinical characteristics and diagnostic methods of intestinal tuberculosis (ITB) in Thailand over 11 years. The length of the manuscript is a bit long. The authors should consider condensing the manuscript to best highlight important findings such as the increase in sensitivity when diagnostic tests were used in combination, the benefit of repeat colonoscopy, and the role of stool tests in the diagnosis of ITB. Numerous grammatical errors in the text should be corrected (suggestions may be found in the reviewed manuscript via tracked changes). Overall, the manuscript had been prepared according to the STROBE checklist except for several items not mentioned in the manuscript:

1) Explanation of how missing data were addressed

2) Reasons why certain tests were not done in all patients

3) Number of missing data for each variable

Below are a few additional revisions that the authors may consider:

1) Abstract: Include all main findings of significance in the conclusion like the increase in sensitivity when diagnostic tests were used in combination and the benefit of repeat colonoscopy for diagnosis.

2) Statistical analysis: Only include statistical analysis that was done. Comparisons of variables with T-test or Chi-square were not seen in results but mentioned, please clarify. The paired T-test should be done instead of McNemar’s test to compare sensitivities as sensitivities are presented as categorical variables.

3) Results: Authors should explain why IGRA, abdominal CT, PCR and culture of colonoscopy specimens, and stool tests were not performed in all patients and the rationale/consideration behind these decisions. Please define the cut-off for positive IGRA results and indicate any risk factors associated with the negative IGRA result. Explain why histopathologic specimens were only available for review in 51 out of 108 patients. Please define “positive tissue pathology” on page 8.

4) Discussion: On page 12, the authors mentioned that PCR only detects the IS6110 gene meanwhile in methods it was mentioned that multiplex PCR was done. Multiplex PCR amplifies more than one target gene. Please clarify.

5) Conclusion: The authors should refrain from mentioning patients requiring empirical treatment as they were not discussed in the results section.

6) Tables and figure legends: It might be better for authors to present table 1 in order of the highest percentage to the lowest for each category. Table 1, colonoscopy findings, number of ulcers: Please define what is meant by multiple ulcers and why patients with 2-5 ulcers and >5 ulcers were not categorized as having multiple ulcers. Table 1, CT findings, abdominal lymphadenopathy: Please current the type “611.3%” and include a description of lymph nodes for the other 18 (39.2%) patients. Figure legend 1: Please add a more specific description for each subfigure 1A-1D.

7) References: Duplicate references were found (reference numbers 10 and 17 also reference numbers 11 and 18), please make necessary changes. Authors should also consider using more recent references as 24 references cited were published more than 10 years ago and 13 references cited were published more than 5 years ago.

Reviewer #3: This is only retrospective study from one tertiary center not large population study

Analysis for sensitivity may not appropriate given the data and not gold standard, the diagnosis criteria intestinal lesion with other organ active TB is also not reliable

The term "Diagnostic yield" more appropriate

6. PLOS authors have the option to publish the peer review history of their article (what does this mean?). If published, this will include your full peer review and any attached files.

Reviewer #1: **Yes: **Mikhail Reshetnikov

Reviewer #2: No

Reviewer #3: No

---

## [Author Response · Author response to Decision Letter 0]

19 Jan 2023

Dear Editor 

We would like to thank the editor and reviewers for examining our manuscript entitled, Clinical characteristics and diagnosis of intestinal tuberculosis in clinical practice at Thailand’s largest national tertiary referral center: An 11-year retrospective review. We appreciate the comments and suggestions, which helped us improve our manuscript. We have reviewed the reviewer's comments and submitted responses as listed below:

Reviewer #1: 

1. Why were patients with perforation of tuberculous intestinal ulcers and intestinal obstruction included in the study?

We included those patients because we aimed to describe the clinical presentations of all patients with intestinal tuberculosis in our center as we showed in Table 1. 

2. How many patients with pulmonary tuberculosis were found to have MBT in sputum?

Of 77 patients with pulmonary involvement, 56 had sputum tested for MTB. The remaining were defined to have pulmonary involvement based on abnormal radiologic images. Of 56 patients with pulmonary involvement who had sputum tests for mycobacteria, 51 had positive results. We have added this information to the first paragraph of the Results section. 

3. What was the level of CD-lymphocytes in patients with HIV infection? How many of them received antiretroviral therapy?

The median CD4 count was 79 cells/mm3 (range 3 – 880 cells/mm3). There were 10 patients receiving antiretroviral therapy before the diagnosis of intestinal tuberculosis. We have added this information to Table 1. 

4. Figure 1C does not respond to intestinal tuberculosis. More consistent with Crohn's disease. What phase of the statement is shown in the figure?

All pictures are from the same patient, and this patient completely responded to anti-tuberculosis treatment. The phase was the venous phase. We added the information in the picture legend. 

5. The statement that abdominal lymphadenopathy in patients amounted to 611.3% (61,3% mistake in main document) is questionable.

Thank you very much for your careful review. We have corrected this typo in Table 1. 

6. In table 2, the sensitivity is calculated incorrectly. These are just descriptive statistics.

We have reviewed the calculated sensitivity in Table 2 and found that all values are correct. 

Reviewer #2: 

The authors report a single institution retrospective review of clinical characteristics and diagnostic methods of intestinal tuberculosis (ITB) in Thailand over 11 years. The length of the manuscript is a bit long. The authors should consider condensing the manuscript to best highlight important findings such as the increase in sensitivity when diagnostic tests were used in combination, the benefit of repeat colonoscopy, and the role of stool tests in the diagnosis of ITB. Numerous grammatical errors in the text should be corrected (suggestions may be found in the reviewed manuscript via tracked changes). Overall, the manuscript had been prepared according to the STROBE checklist except for several items not mentioned in the manuscript:

1) Explanation of how missing data were addressed

2) Reasons why certain tests were not done in all patients

3) Number of missing data for each variable

Thank you very much for your insightful suggestion. We decided to include demographic data and clinical manifestations in our manuscript because we thought that these details were useful for medical students and general practitioners. Therefore, we leave these details in the results. However, we shortened the main text as per the reviewer’s suggestion and provided all details in Table 1. 

Regarding the missing data, we addressed this issue by not including the missing data in the analysis of those variables. For example, the descriptive findings of endoscopic findings, CT findings, and pathological findings were calculated from only those who had the data available. 

The reasons why certain diagnostic tests were not done in all patients are due to the individual decision of the treating physicians and the availability of the tests at that time. This issue is an important limitation of this retrospective study. Therefore, a prospective study is warranted. We have emphasized this issue in the discussion section. 

The number of missing data for each variable can be calculated back from the number of available data, which has been noted in the manuscript for all variables. 

Below are a few additional revisions that the authors may consider:

1) Abstract: Include all main findings of significance in the conclusion like the increase in sensitivity when diagnostic tests were used in combination and the benefit of repeat colonoscopy for diagnosis.

We have added these significant findings in the conclusion as suggested. 

2) Statistical analysis: Only include statistical analysis that was done. Comparisons of variables with T-test or Chi-square were not seen in results but mentioned, please clarify. The paired T-test should be done instead of McNemar’s test to compare sensitivities as sensitivities are presented as categorical variables.

Thank you very much for your careful review. We have deleted the statistics which were not used in the manuscript. For the comparison of the sensitivity of the two tests in the same patients, we decided to use McNemar’s test. The paired T-test should be used for the comparison of continuous variables in the same patients. 

3) Results: Authors should explain why IGRA, abdominal CT, PCR and culture of colonoscopy specimens, and stool tests were not performed in all patients and the rationale/consideration behind these decisions. Please define the cut-off for positive IGRA results and indicate any risk factors associated with the negative IGRA result. Explain why histopathologic specimens were only available for review in 51 out of 108 patients. Please define “positive tissue pathology” on page 8.

Each diagnostic test was sent depending on the individual decision of the treating physicians and the availability of the tests at that time. The stool tests were preferred in patients with immunocompromised status, particularly those with HIV infection; this was possibly due to avoiding the risk of infectious contamination during the procedure. 

Interferon-gamma release assay for tuberculosis was measured using Quantiferon-TB Gold Plus (Qiagen, Dusseldorf, Germany), a Sandwich enzyme-linked immunosorbent assay. The cutoff value was 0.35 IU/ml. The risk factors associated with the negative IGRA result include advanced age, low peripheral lymphocyte counts, and immunosuppressive conditions. We have added this information to the Methods section, subsection “Laboratory investigations for ITB”. 

The histopathologic specimens were only available for review in 51 out of 108 patients because some specimens were not kept and some specimens had the stain fading out, resulting in an inability to be reviewed. 

The positive tissue histopathology referred to either presence of AFB or caseous necrosis in the tissue specimen. The definition was in the parenthesis in the sentence “As shown in Table 2, the sensitivity of tissue AFB, caseous necrosis, and pathology (presence of either AFB or caseous necrosis) was 40.7% (44/108), 13.9% (15/108), and 47.2% (51/108), respectively.” in the first paragraph of the subsection “Diagnosis of intestinal tuberculosis by colonoscopy.” 

4) Discussion: On page 12, the authors mentioned that PCR only detects the IS6110 gene meanwhile in methods it was mentioned that multiplex PCR was done. Multiplex PCR amplifies more than one target gene. Please clarify.

Thank you very much for your careful review. We have discussed with the microbiology lab of our hospital and found that the test used in our hospital can detect several TB gene targets based on automated and semi-automated kits for mycobacterial DNA detection, not only the IS6110 gene. Therefore, we deleted this issue from the manuscript. 

5) Conclusion: The authors should refrain from mentioning patients requiring empirical treatment as they were not discussed in the results section.

We have revised the conclusion to “Despite the availability of PCR and culture for TB, the overall diagnostic sensitivity was found to be low in this cohort. The sensitivity increased when these tests were used in combinations. Repeated colonoscopy may be beneficial. Adding stool mycobacteria tests did not significantly increase the diagnostic yield if endoscopy was performed, but it could be beneficial if endoscopy is unfeasible.” 

6) Tables and figure legends: It might be better for authors to present table 1 in order of the highest percentage to the lowest for each category. Table 1, colonoscopy findings, number of ulcers: Please define what is meant by multiple ulcers and why patients with 2-5 ulcers and >5 ulcers were not categorized as having multiple ulcers. Table 1, CT findings, abdominal lymphadenopathy: Please current the type “611.3%” and include a description of lymph nodes for the other 18 (39.2%) patients. Figure legend 1: Please add a more specific description for each subfigure 1A-1D.

We have revised Table 1 as suggested. We categorized the clinical presentations into “Gastrointestinal symptoms”, “Systemic symptoms”, “Pulmonary involvement”, and “Lymphadenopathy”, and reordered the findings from the highest percentage to the lowest percentage for each category. 

For the endoscopic findings, the number of ulcers was reviewed from the official endoscopic reports. Some reports provided exact numbers of ulcers while some provided the detail roughly (multiple ulcers). This is the reason why the number of ulcers was categorized as “1”, “2-5”, “>5”, and “multiple” in our manuscript. However, we agree that presenting this data like this could be confusing. So, we decided to change the category of ulcer numbers to “Single ulcer” and “Multiple ulcers”. Furthermore, 6 patients with missing data were categorized as “Unknown” because there was no detail about the number of ulcers in their official endoscopic reports. 

For CT findings, we categorized the findings into “Bowel wall findings”, “Involved pattern”, “mesenteric findings”, “abdominal lymphadenopathy”, and “ascites”. A description of lymph nodes for the other 18 patients was “Enlarged lymph node without necrosis and calcification”. The typo “611.3%” was corrected. 

For Figure 1 legend, we added the information as follows: “Fig 1. Endoscopic (Fig 1A and 1B), computed tomography (Fig 1C), and pathological findings (Fig 1D) from the patient with intestinal tuberculosis. Fig1A and 1B showed a large ulcer involving the IC valve. Fig1C was an axial contrast-enhanced CT scan showing moderate segmental wall thickening with increased mucosal and mural enhancement at the terminal ileum with associated increasing IC valve thickness. Fig 1D showed ulcerated ileal mucosa with dense chronic inflammatory cell infiltration and aggregates of epithelioid histiocytes (granuloma) with central necrosis. (arrow)."

7) References: Duplicate references were found (reference numbers 10 and 17 also reference numbers 11 and 18), please make necessary changes. Authors should also consider using more recent references as 24 references cited were published more than 10 years ago and 13 references cited were published more than 5 years ago.

Thank you very much. We have corrected the duplicated references and added some more recently published articles as follows: 

1. Kang W, Yu J, Du J, Yang S, Chen H, Liu J, et al. The epidemiology of extrapulmonary tuberculosis in China: A large-scale multi-center observational study. PloS one. 2020;15(8):e0237753. Epub 2020/08/22. doi: 10.1371/journal.pone.0237753. PubMed PMID: 32822367; PubMed Central PMCID: PMCPMC7446809.

2. Khan MK, Islam MN, Ferdous J, Alam MM. An Overview on Epidemiology of Tuberculosis. Mymensingh medical journal : MMJ. 2019;28(1):259-66. Epub 2019/02/14. PubMed PMID: 30755580.

3. Gaifer Z. Epidemiology of extrapulmonary and disseminated tuberculosis in a tertiary care center in Oman. International journal of mycobacteriology. 2017;6(2):162-6. Epub 2017/06/01. doi: 10.4103/ijmy.ijmy_31_17. PubMed PMID: 28559518.

4. Sama JN, Chida N, Polan RM, Nuzzo J, Page K, Shah M. High proportion of extrapulmonary tuberculosis in a low prevalence setting: a retrospective cohort study. Public health. 2016;138:101-7. Epub 2016/05/04. doi: 10.1016/j.puhe.2016.03.033. PubMed PMID: 27137870; PubMed Central PMCID: PMCPMC5012930.

5. Agarwala R, Singh AK, Shah J, Mandavdhare HS, Sharma V. Ileocecal thickening: Clinical approach to a common problem. JGH open : an open access journal of gastroenterology and hepatology. 2019;3(6):456-63. Epub 2019/12/14. doi: 10.1002/jgh3.12186. PubMed PMID: 31832544; PubMed Central PMCID: PMCPMC6891021.

6. Paulose RR, Kumar VA, Sharma A, Damle A, Saikumar D, Sudhakar A, et al. An outcome-based composite approach for the diagnosis of intestinal tuberculosis: a pilot study from a tertiary care centre in South India. The journal of the Royal College of Physicians of Edinburgh. 2021;51(4):344-50. Epub 2021/12/10. doi: 10.4997/jrcpe.2021.406. PubMed PMID: 34882131.

7. Pratap Mouli V, Munot K, Ananthakrishnan A, Kedia S, Addagalla S, Garg SK, et al. Endoscopic and clinical responses to anti-tubercular therapy can differentiate intestinal tuberculosis from Crohn's disease. Alimentary pharmacology & therapeutics. 2017;45(1):27-36. Epub 2016/11/05. doi: 10.1111/apt.13840. PubMed PMID: 27813111.

8. Kedia S, Das P, Madhusudhan KS, Dattagupta S, Sharma R, Sahni P, et al. Differentiating Crohn's disease from intestinal tuberculosis. World journal of gastroenterology. 2019;25(4):418-32. Epub 2019/02/01. doi: 10.3748/wjg.v25.i4.418. PubMed PMID: 30700939; PubMed Central PMCID: PMCPMC6350172.

9. Udgirkar S, Jain S, Pawar S, Chandnani S, Contractor Q, Rathi P. CLINICAL PROFILE, DRUG RESISTANCE PATTERN AND TREATMENT OUTCOMES OF ABDOMINAL TUBERCULOSIS PATIENTS IN WESTERN INDIA. Arquivos de gastroenterologia. 2019;56(2):178-83. Epub 2019/08/29. doi: 10.1590/s0004-2803.201900000-35. PubMed PMID: 31460583.

10. Weledji EP, Pokam BT. Abdominal tuberculosis: Is there a role for surgery? World journal of gastrointestinal surgery. 2017;9(8):174-81. Epub 2017/09/22. doi: 10.4240/wjgs.v9.i8.174. PubMed PMID: 28932351; PubMed Central PMCID: PMCPMC5583525.

Reviewer #3: 

This is only retrospective study from one tertiary center not large population study

Analysis for sensitivity may not appropriate given the data and not gold standard, the diagnosis criteria intestinal lesion with other organ active TB is also not reliable. The term "Diagnostic yield" more appropriate

Thank you very much. We have revised the sentence in the discussion section, subsection Strengths and limitations to “First, this study included a substantial number of patients from a single referral center.” 

For the diagnostic criteria, we agree that the intestinal lesion with other organs active TB could result from causes other than ITB. However, the diagnosis of ITB was confirmed by responding to anti-TB treatment. Therefore, we decided to keep the term “sensitivity” in the manuscript. 

Thank you very much for all your valuable and insightful suggestions. Our manuscript has been much improved. 

We are submitting our revised manuscript for consideration and hope you will agree that our study is of enough scientific relevance to justify publication in PLOS ONE after the peer-review process. 

Yours sincerely, 

Julajak Limsrivilai, MD, MSc

Associate Professor of Medicine

Siriraj Hospital, Mahidol University, Bangkok, Thailand

---

## [Decision Letter · Decision Letter 1]

14 Feb 2023

Clinical characteristics and diagnosis of intestinal tuberculosis in clinical practice at Thailand’s largest national tertiary referral center: An 11-year retrospective review

PONE-D-22-30114R1

Dear Dr. Limsrivilai,

We’re pleased to inform you that your manuscript has been judged scientifically suitable for publication and will be formally accepted for publication once it meets all outstanding technical requirements.

Kind regards,

Mao-Shui Wang

Academic Editor

PLOS ONE

Additional Editor Comments (optional):

Reviewers' comments:

Reviewer's Responses to Questions

**Comments to the Author**

1. If the authors have adequately addressed your comments raised in a previous round of review and you feel that this manuscript is now acceptable for publication, you may indicate that here to bypass the “Comments to the Author” section, enter your conflict of interest statement in the “Confidential to Editor” section, and submit your "Accept" recommendation.

Reviewer #1: (No Response)

Reviewer #3: All comments have been addressed

2. Is the manuscript technically sound, and do the data support the conclusions?

Reviewer #1: Yes

Reviewer #3: Yes

3. Has the statistical analysis been performed appropriately and rigorously? 

Reviewer #1: I Don't Know

Reviewer #3: Yes

4. Have the authors made all data underlying the findings in their manuscript fully available?

Reviewer #1: Yes

Reviewer #3: Yes

5. Is the manuscript presented in an intelligible fashion and written in standard English?

Reviewer #1: (No Response)

Reviewer #3: Yes

6. Review Comments to the Author

Reviewer #1: (No Response)

Reviewer #3: (No Response)

7. PLOS authors have the option to publish the peer review history of their article (what does this mean?). If published, this will include your full peer review and any attached files.

Reviewer #1: **Yes: **MIKHAIL NIKOLAEVICH RESHETNIKOV

Reviewer #3: **Yes: **hasan maulahela

---

## [Editor Report · Acceptance letter]

5 Apr 2023

PONE-D-22-30114R1 

Clinical characteristics and diagnosis of intestinal tuberculosis in clinical practice at Thailand's largest national tertiary referral center: An 11-year retrospective review 

Dear Dr. Limsrivilai:

I'm pleased to inform you that your manuscript has been deemed suitable for publication in PLOS ONE. Congratulations! Your manuscript is now with our production department. 

Kind regards, 

on behalf of

Dr. Mao-Shui Wang 

Academic Editor

PLOS ONE